# The role of land-atmosphere coupling in subseasonal surface air temperature prediction across the contiguous United States

Yuna Lim[1,2], Andrea M. Molod[2], Randal D. Koster[2], Joseph A. Santanello[3]

[1]Earth System Science Interdisciplinary Center, University of Maryland College Park, College Park, 20740, USA
[2]Global Modeling and Assimilation Office, NASA Goddard Space Flight Center, Greenbelt, Maryland, 20771, USA
[3]Hydrological Sciences Laboratory, NASA Goddard Space Flight Center, Greenbelt, Maryland, 20771, USA

*Correspondence to*: Yuna Lim (yu-na.lim@nasa.gov)

**Abstract.** Land-atmosphere (L-A) coupling can play a crucial role for subseasonal-to-seasonal (S2S) predictability and prediction. When coupling is strong, L-A processes and feedback are expected to enhance the system's memory, thereby
increasing the predictability and prediction skill. This study evaluates subseasonal prediction of ambient surface air temperature under conditions of strong versus weak L-A coupling in forecasts produced with NASA's state-of-the-art Goddard Earth Observing System (GEOS) S2S forecast system. By applying three L-A coupling metrics that collectively capture the connection between the soil and the free troposphere, we observe improved prediction skill for surface air temperature during weeks 3-4 of boreal summer forecasts across the Midwest and northern Great Plains, particularly when all three indices indicate
strong L-A coupling at this lead time. The prediction skill indeed increases as more indices show strong coupling. The forecasts with strong L-A coupling in these regions tend to exhibit sustained warm and dry anomalies, signals that are well simulated in the model. Overall, this study highlights how better identifying and capturing relevant L-A coupling processes can potentially enhance prediction on S2S timescales.

## 1 Introduction

Subseasonal-to-seasonal prediction, involving forecasts covering a period of 15 to 60 days ahead, holds significance for end users in decision-making roles whose goal is to mitigate human and economic losses caused by natural disasters. In this context, the S2S prediction project was initiated by the World Weather Research Programme / World Climate Research Programme, with operational centers worldwide providing their retrospective forecasts to the project to advance the understanding of the S2S prediction (Vitart et al., 2017). Factors contributing to subseasonal prediction include soil moisture anomalies and their
associated memory, the Madden-Julian oscillation, and sudden stratospheric warming (e.g. Dirmeyer et al., 2018; Lim et al., 2018; Domeisen et al., 2020). However, because such factors fall far short of providing perfect prediction skill, finding "forecasts of opportunity" stemming from these factors can optimize the utilization of a given prediction system (Mariotti et al., 2020).

Land-atmosphere coupling is required for soil moisture to affect surface meteorological conditions, which in turn can amplify
the memory of the system (Seneviratne et al., 2010). It occurs across various timescales and involves numerous variables

(Seneviratne et al., 2010; Santanello et al., 2018) including land surface and planetary boundary layer (PBL) states and fluxes. Positive feedback in L-A coupling involves soil moisture, surface latent and sensible heat fluxes, surface temperature, boundary layer temperature and humidity, and cloud formation (see Fig. 6 of Molod et al., 2004 for one example of a positive feedback loop). Such robust L-A coupling processes can significantly impact droughts and heat extremes (Fisher et al., 2007; Dirmeyer et al., 2021). Of course, to be impactful, the strength of L-A coupling must exert a stronger influence on the local conditions than advection from synoptic-scale atmospheric motion.

Accurate representation of strong L-A coupling is necessary to leverage the impact of soil moisture on surface meteorological variables in a prediction system (Roundy et al., 2014; Benson and Dirmeyer, 2023). Previous studies have investigated how prediction models simulate L-A coupling, examining the connections between variables (e.g., Dirmeyer, 2013; Abdolghafoorian and Dirmeyer, 2021) or assessing the simulation of the relationship between soil moisture and evaporation (Benson and Dirmeyer, 2023). It has been observed that the ability to simulate L-A coupling influences the prediction of surface meteorological variables. For example, accurate simulation of soil moisture "breakpoints", which is a critical threshold value indicating when atmosphere responses to the land surface conditions occur, can improve predictions of hot extremes (Benson and Dirmeyer, 2023). Roundy et al. (2014) demonstrated that coupling strength is closely associated with predictions of surface temperature and precipitation, based on two drought events. However, in terms of "forecasts of opportunity", no attempts have yet been made to assess the impact of L-A coupling strength on the prediction skill of surface variables at these subseasonal timescales.

In this study, we hypothesize that the surface air temperature forecasts for weeks 3-4 will be better predicted if conditions conducive to positive L-A feedback are present in that forecast. Furthermore, we aim to assess the importance of fully integrating the soil moisture-atmosphere connection by considering separately the links between soil moisture and evapotranspiration (ET), between ET and surface skin temperature, and between ET and the character of the PBL and free troposphere, hypothesizing that all three links must be strong for the feedback to positively affect prediction skill. The analysis focuses on the contiguous United States, which contains a well-known hotspot of L-A coupling (e.g., Koster et al., 2006). Importantly, this approach focuses specifically on forecasts of these coupling components, rather than on observations and/or the forecast initialization (as is typical in prediction studies), when evaluating impacts on forecast skill.

This paper is organized as follows. Section 2 describes the forecast system used in our analysis, the reanalysis data used for assessing the prediction skill, the metrics we use to quantify L-A coupling strength and the evaluation metrics of prediction skill. Section 3 presents the results, including the general characteristics of L-A coupling as simulated by the model and the impacts of strong versus weak coupling on surface air temperature prediction skill. Summary and additional discussion are provided in Sect. 4.

## 2 Data and Methods

### 2.1 GEOS-S2S Model

In this study, we used the suite of subseasonal forecasts with Version 2 of National Aeronautics and Space Administration (NASA)'s state-of-the-art GEOS S2S analysis and forecast system, i.e., GEOS-S2S-2 (Molod et al., 2020). This system comprises a coupled atmosphere-ocean General Circulation Model (GCM) and a weakly coupled data assimilation system. The coupled atmosphere-ocean GCM incorporates several components: the GEOS atmospheric GCM (after Molod et al., 2015 and Reinecker et al., 2008), Modular Ocean Model version 5 (MOM5) ocean GCM (Griffies, 2012), the GOCART aerosol model (Chin et al., 2002), and the Community Ice CodE-4 sea ice model (Hunke, 2008). The atmospheric and oceanic components have approximately 0.5°x0.5° horizontal resolution with 72 vertical levels for the atmosphere and 40 for the ocean. Land processes are represented using the Catchment model (Ducharne et al., 2000; Koster et al., 2000), a well-established land model with a non-traditional representation of subgrid hydrological variability. With the Catchment model, the land surface is divided into a patchwork of irregularly-shaped hydrological catchments, and each catchment element is in turn divided into three distinct subareas (saturated, subsaturated but transpiring, and wilting) with sizes that vary in time based on the model's bulk soil moisture prognostic variables. Distinct physical parameterizations for evapotranspiration and runoff are utilized in the different subareas.

The weakly coupled data assimilation system used for the forecast system's initialization makes use of observed (rain gauge-based) precipitation measurements to drive the land surface fields; as a result, the soil moisture initialization for the forecasts appropriately reflects the character of the antecedent observed precipitation. Quantitative estimates of the accuracy of such precipitation-driven, model-based estimates are rare but do exist. Using a slightly modified version of the land model used here in GEOS (Catchment model), Reichle et al. (2017) found strong temporal correlations between the surface and root zone soil moisture so generated and corresponding in-situ measurements across the contiguous US (CONUS). The soil moistures produced by a slightly updated version of the land model were recently compared to Soil Moisture Active Passive (SMAP) satellite-based soil moisture retrievals (Entekhabi et al., 2009); the resulting anomaly correlation coefficients over CONUS were found to generally exceed 0.6 (Qing Liu, personal comm., 2024). Koster et al. (2020, their Figure 9) effectively show that significant subseasonal air temperature forecast skill in the GEOS-S2S-2 system, particularly in the eastern half of CONUS, is strongly tied to the initialized profile soil moisture, indicating useful accuracy therein.

This study utilizes all available retrospective and real-time forecasts spanning from 1999 to 2022, except for the summer of 2017, which is the transition period between the retrospective and real-time forecasts. The full forecast suite is the combination of the retrospective and real-time forecasts, which will be referred to as forecasts hereafter. Each forecast includes four ensemble members and is initialized six times a month at 5-day intervals. Since the L-A coupling is the strongest in the boreal summer (e.g., Koster et al., 2004), we examined the forecasts initialized in June-August, resulting in 18 start dates per year. A total of 414 forecast dates are thus analyzed in this study. Note that the forecasts initialized in late August provide forecast information extending into September.

## 2.2 Verification Data

To assess forecast skill, we primarily take as "observations" the 2-meter temperatures (T2m) produced in the 5th generation of the European Centre for Medium-Range Weather Forecast Reanalysis (ERA5; Hersbach et al., 2020). During the production of ERA5, the data assimilation process ingested substantial amounts of T2m information from ground stations, making the ERA5 T2m product particularly trustworthy as an evaluation standard, particularly in boreal summer in the absence of snow cover. We aggregated the hourly 0.25°x0.25° ERA5 T2m data during the period 1999-2022 into daily means for consistency with the available GEOS-S2S-2 forecast data. In addition to T2m, we used other variables such as surface soil moisture, surface latent heat flux, surface skin temperature, 2-meter specific humidity (Q2m), and PBL height (PBLH) for evaluating L-A coupling processes in the model. As an additional dataset for evaluation, we utilized data from a second reanalysis, the Modern-Era Retrospective analysis for Research and Applications, version 2 (MERRA-2; Gelaro et al., 2017); we aggregated the hourly 0.625°x0.5° MERRA-2 data from 1999 to 2022 into daily means. Due to slight differences in grid resolution across the ERA5, GEOS-S2S-2, and MERRA-2, all datasets were interpolated to the same 1.0°x1.0° grid before analysis.

## 2.3 Land-Atmosphere Coupling Metrics

L-A coupling strength has often been discussed in terms of "two-legged metrics" (e.g., Dirmeyer, 2011, Santanello et al., 2018), statistical metrics that separately describe the connection between land conditions and the surface fluxes, as well as the connection between the surface fluxes and the overlying atmosphere or boundary layer. By dividing the coupling process into these two components, it becomes easier to identify the complex processes governing these feedback processes. Specifically, for the first metric of L-A coupling to be used in this study we calculate the correlation between surface soil moisture (SM) and surface latent heat flux (LH) using Eq. (1). A higher positive correlation signifies a more robust L-A coupling between these two variables, indicating that the soil moisture anomalies are driving the latent heat anomalies. The second metric to be used here is the correlation between LH and surface skin temperature (TS), (Eq. 2). Here, a greater negative correlation between LH and TS signifies a stronger L-A coupling, indicating that the latent heat anomalies are driving the skin temperature anomalies. To detect changes in conditions conducive to clouds and convection formation, for the third metric we compute the correlation between the anomaly of the lifting condensation level deficit (LCLd or LCL deficit; Santanello et al., 2011) and the anomaly of LH (Eq. 3). The LCLd is defined here as the difference in meters between the PBLH and the LCL, where a negative value indicates that the LCL has not been reached by the air parcel subjected to PBL turbulent motions, meaning that the conditions are not favorable to allow convection and perhaps precipitation. LCLd anomalies of either sign, therefore, are associated with conditions that are more or less favorable to convection. The LCL is estimated from T2m and Q2m (Bolton, 1980), while PBLH is provided by the forecasts. A higher positive correlation of LH and LCLd, where positive (negative) LH anomalies correspond to positive (negative) LCLd anomalies, therefore describes a more intense L-A coupling.

As mentioned in the introduction, our focus lies on conditions for which the forecast produces a strong L-A coupling. In line with this, each index is computed using data from the forecasts rather than from the reanalysis. In line with our focus on

subseasonal prediction, each index at a given grid point is calculated using data from weeks 3-4 (days 15 to 28) of each of the forecast's four ensemble members, as expressed below:

$$r(SM, LH) = \frac{\sum_{k=1}^{4}\sum_{\tau=15}^{28} SM'LH'}{\sqrt{\sum_{k=1}^{4}\sum_{\tau=15}^{28} SM'^2}\sqrt{\sum_{k=1}^{4}\sum_{\tau=15}^{28} LH'^2}} , \tag{1}$$

$$r(LH, TS) = \frac{\sum_{k=1}^{4}\sum_{\tau=15}^{28} LH'TS'}{\sqrt{\sum_{k=1}^{4}\sum_{\tau=15}^{28} LH'^2}\sqrt{\sum_{k=1}^{4}\sum_{\tau=15}^{28} TS'^2}} , \tag{2}$$

$$r(LH, LCLd) = \frac{\sum_{k=1}^{4}\sum_{\tau=15}^{28} LH'LCLd'}{\sqrt{\sum_{k=1}^{4}\sum_{\tau=15}^{28} LH'^2}\sqrt{\sum_{k=1}^{4}\sum_{\tau=15}^{28} LCLd'^2}} , \tag{3}$$

where $k$ is the ensemble number, and $\tau$ is the forecast lead time. As indicated by the primes, corresponding lead time dependent climatological values are subtracted from all variables before computing the index.

For a given index, separate values are computed for each forecast, and then the values are ranked into percentiles. If a given forecast places an index in the upper 50th percentile, that index is categorized "strong" for the forecast. When all three indices

indicate strong (weak) L-A coupling, it is classified as a strong (weak) L-A coupling event. This approach ensures that the full process chain connecting soil moisture, surface energy budget, and boundary layer remains 'active'. Later in this paper, we will also examine forecasts for which only one or two indices indicate strong L-A coupling.

**2.4 Evaluation Metrics**

The anomaly correlation coefficient (ACC) between GEOS-S2S-2 and ERA5 is computed to quantify the prediction skill. The

140 ACC at 3-4 weeks lead can be expressed as:

$$ACC = \frac{\sum_{t=1}^{N}(f(t)-\bar{f})(O(t)-\bar{O})}{\sqrt{\sum_{t=1}^{N}(f(t)-\bar{f})^2}\sqrt{\sum_{t=1}^{N}(O(t)-\bar{O})^2}} , \tag{4}$$

where $f(t)$ is the forecast datum at weeks 3-4 lead for the forecast initialized on date $t$, and $O(t)$ is obtained from ERA5 for the corresponding period. More precisely, $f(t)$ and $O(t)$ are the ensemble-mean forecast T2m averaged over weeks 3-4 (i.e., days 15-28) of the forecast and the ERA5 T2m averaged over the same period, respectively. $\bar{f}$ and $\bar{O}$ are the climatological

values for the forecast and ERA5 data, derived (for the forecasts) by averaging the data initialized on the same start dates. By subtracting the climatological means from the model and the observational data, impacts of the seasonal cycle, model drift, and mean bias on the calculated ACC can be excluded or at least mitigated. $N$ is the number of initialization dates used in the analysis.

## 3 Results

Figure 1 reveals the character of the simulated L-A coupling in GEOS-S2S-2 and in the two sets of reanalysis data. The L-A coupling index is computed for a given forecast at a week 3-4 forecast lead time using Eqs. (1)-(3) above and, for the figure, is averaged over forecasts initialized between June 1999 to August 2022 (Sect. 2.3). The coupling index for either reanalysis is computed using data from corresponding days. Generally, the datasets reveal similar spatial distributions in all three indices. The maximum (in magnitude) correlations are found in the western U.S., with notable asymmetries between the western and eastern U.S. Positive values of r(SM, LH) are broadly evident across the continental U.S., with distinctively high values observed in the deserts of the western U.S. and Great Plains extending from Mexico to North Dakota – locations characterized by a soil moisture-controlled evaporative regime, as independently mapped with satellite-based evapotranspiration and soil moisture estimates (Koster et al., 2024). These positive r(SM, LH) values indicate that higher soil moisture leads to increased evapotranspiration, which should in turn be related to lower surface temperatures through evaporative cooling, which explains why the r(SM, LH) patterns are largely aligned (though opposite in sign) with those for r(LH, TS). The r(LH, LCLd) patterns are also similar to those for r(LH, TS), again with an opposite sign. As the latent heat flux increases, both relative humidity and dew point temperature increase, causing a decrease in the LCL (Seo and Dirmeyer, 2022) and a consequent increase in LCLd (i.e., PBLH - LCL height in meters). Note that the LCL is more sensitive to changes in latent heat flux while the PBLH is generally driven by surface heating and buoyancy flux, resulting in positive r(LH, LCLd) in the western U.S. In the western and southeastern U.S., the average coupling strength of GEOS-S2S-2 falls between that of ERA5 and MERRA-2.

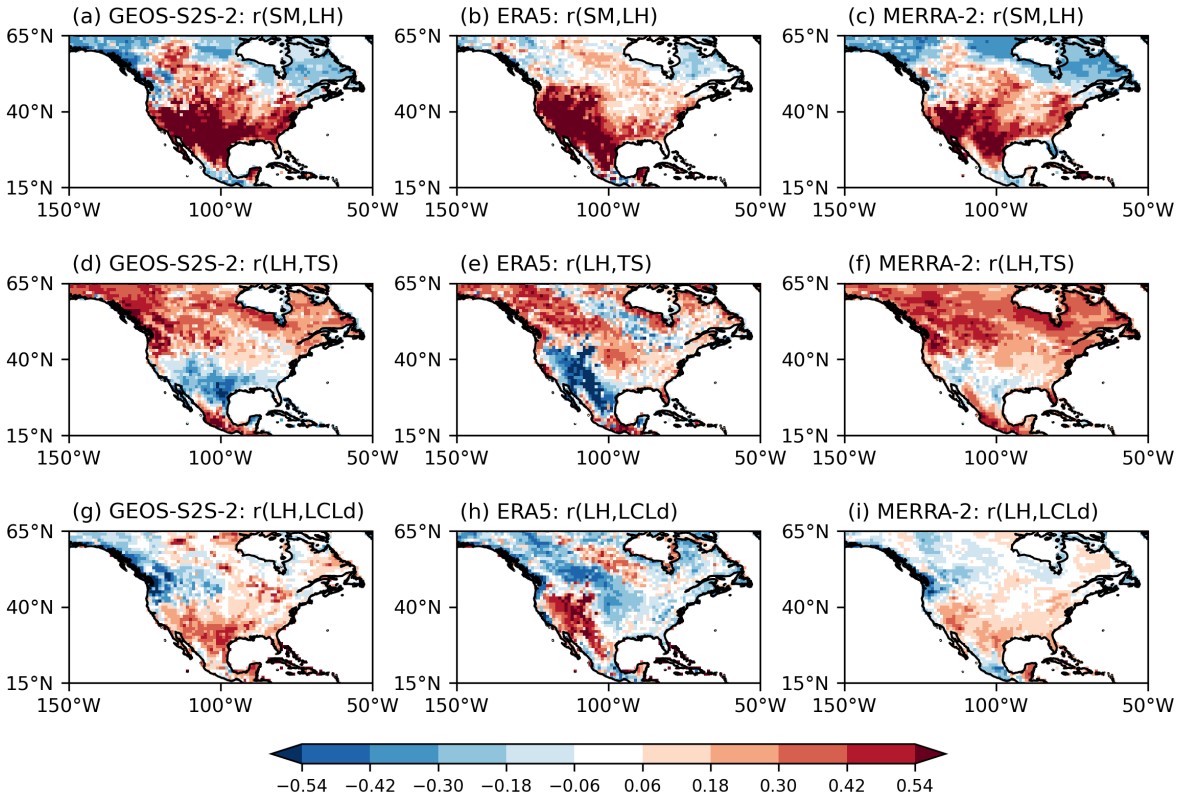

**Figure 1: (a) Spatial distribution of the r(SM, LH) averaged across all forecasts initialized from 1999 to 2022. (b-c) Same as (a), but derived from ERA5 and MERRA-2, respectively. (d-i) Same as (a-c) but the spatial distributions for (d-f) the r(LH, TS) and (g-i) the r(LH, LCLd).**

The spatial distribution of GEOS-S2S-2 is understandably more similar to that of MERRA-2, as both are based on the same modeling system (Molod et al., 2020). The asymmetry between the western and eastern U.S. is somewhat more pronounced in ERA5 compared to GEOS-S2S-2 and MERRA-2. As mentioned earlier, the data assimilation process used in the production of ERA5 incorporated substantial amounts of T2m information from ground stations. Furthermore, ERA5 is based on higher spatial resolution: ERA5 data are provided at approximately 25 km horizontal resolution, whereas MERRA-2 provides at 50 km. This difference may contribute to variations in simulating boundary layer conditions and land-atmosphere coupling. Nevertheless, using MERRA-2 is still valuable for examining the sensitivity of the results to the choice of reanalysis data. Some disparities are found in the eastern U.S., with the average coupling strength of GEOS-S2S-2 slightly higher than that of both reanalysis datasets. These disparities can perhaps be attributed to data assimilation process that accompanies the model during reanalysis, which enforces observed values into the reanalysis data and may suppress or enhance physical correlations and L-A feedback modeled by the system. These considerations would not, of course, impact the forecasts made during the

forecast period. The discrepancy between MERRA-2 and GEOS-S2S-2 particularly supports this explanation, given that the atmospheric and land components of the two systems are structurally similar.

Figure 2a presents the ACC of T2m for week 3-4 forecasts throughout the entire analysis period (Sect. 2.4). The T2m anomalies in each forecast are averaged over weeks 3-4, and the ACC is computed by comparing these T2m averages with the corresponding 2-week averages obtained from ERA5. Overall prediction skill over the continental U.S. is about 0.3 of ACC. The ACC values are higher in the western, southcentral, and eastern U.S. compared to the northcentral U.S. Overall, GEOS-S2S-2 performs similarly to other models (Wang and Robertson, 2018; Pegion et al., 2019). In the eastern U.S., the ACC for GEOS-S2S-2 is between 0.2 and 0.3, which is slightly below the ECMWF Variable Resolution Ensemble Prediction System monthly forecast system (VarEPS) but exceeds that of NCEP Climate Forecast System, version 2 (CFSv2; see Fig. 2 of Wang and Robertson, 2018). Texas and the southeastern U.S. region have relatively high ACC in GEOS-S2S-2; the values are close to 0.4, which is comparable to ECMWF VarEPS and NCEP CFSv2 (Wang and Robertson, 2018). In the western U.S., the GEOS-S2S-2 shows ACC values above 0.3, slightly outperforming ECMWF VarEPS and NCEP CFSv2 (Wang and Robertson, 2018). Although a direct comparison with the results in Fig. 2a is difficult because Pegion et al. (2019) evaluated the predictability of week 3 forecasts in Subseasonal Experiment (SubX) models, they showed that the GEOS-S2S-2 model has relatively better T2m predictability over the continental U.S. compared to other SubX models.

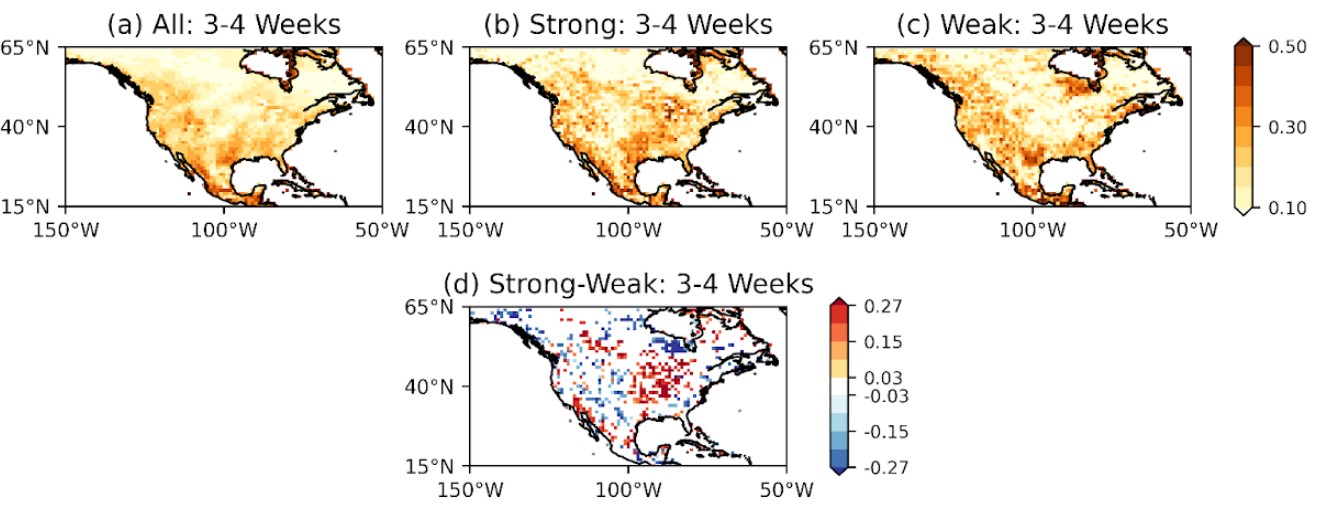

**Figure 2: The ACC of T2m anomalies (forecast values versus observations) at a forecast lead time of week 3-4 over North America during (a) all, (b) strong, and (c) weak L-A coupling events. Strong (weak) events are defined when all three indices are above (below) the 50 percentiles. (d) The difference in T2m ACC between strong and weak events. Statistically significant differences at a 90% confidence level are shaded.**

Figure 2 illustrates how L-A coupling strength affects the ACC of T2m. Based on the 50th percentiles of each L-A coupling index, we identify forecasts characterized by strong and weak L-A couplings. (See Sect. 2.3; all three indices must be in the

upper (lower) 50th percentile for the forecast to be identified as having strong (weak) L-A coupling. Forecasts where not all three indices fall within the same percentile range (about half of the total sample size) are excluded from this particular

analysis.) The ACC of T2m during the strong and weak L-A coupling forecast subsets are depicted in Figs. 2b and c, respectively. A distinct difference in ACC is observed between strong and weak L-A coupling events, particularly across the northern Great Plains, Midwest, and western coast of Mexico, where strong coupling is tied to higher ACC. The strong-weak difference (Fig. 2d) in these areas is maximally up to 0.20 and is statistically significant at a 90% confidence level, as determined by a bootstrapping test. (In the bootstrapping test, the full dataset is subsampled randomly to produce two subsets

of equal size to our strong and weak groups, and then the differences in their skills are computed. This process is repeated 10,000 times. The observed difference is compared to those obtained from the random sampling.) These regions align with the hotspots of strong L-A coupling identified in multimodel analysis (Fig. 8c of Seneviratne et al., 2010). Similar differences were found when analyzing the 30th and 40th percentiles instead of the 50th percentile (not shown). To test whether our spatial interpolation of the GEOS-S2S-2 data had some impact on our findings, we conducted the same analysis (not shown) using a

0.5°x0.5° grid without the spatial interpolation. The results were essentially the same.

The southern Great Plains also exhibit a strong L-A coupling, as can be inferred from Fig. 1, but the prediction skill in GEOS-S2S-2 does not vary with the L-A coupling strength in this region. Considering the fact that the hotspots of L-A coupling can differ between models (Koster et al., 2006), it is possible that T2m in this region is controlled more strongly by external factors (e.g., advection) than by the land component, which could diminish the effect of L-A coupling on prediction skill.

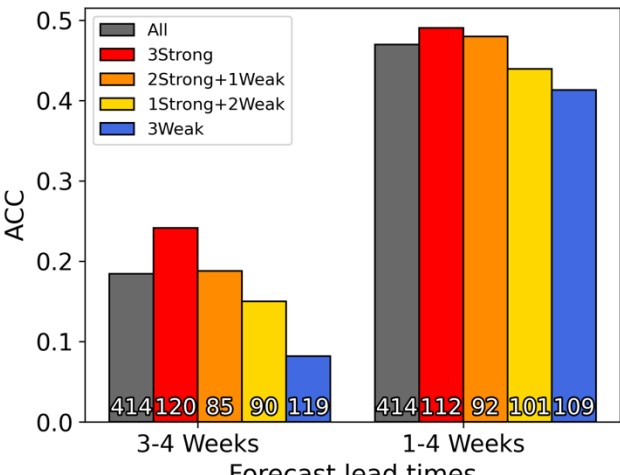

**Figure 3: A bar plot illustrating the T2m ACC averaged across the northern Great Plains (100°-85°W, 35°-45°N) at forecast lead times of 3-4 weeks and 1-4 weeks for each coupling group. Black bar indicates ACC for all forecasts. Red (blue) bars denote ACC values with all three indices strong (weak), while orange (yellow) bars indicate ACC with any two (one) indices strong and any one**
**(two) index weak. The average sample size in each group is indicated at the bottom of each bar.**

Focusing on the region exhibiting the most pronounced impact of L-A coupling, we examine whether incorporating more elements of the L-A coupling feedback loop enhances the prediction skill. Figure 3 displays the ACC of T2m averaged over the northern Great Plains for four different and mutually exclusive forecast subsets. The 3Strong and 3Weak groups correspond

to the strong and weak L-A coupling groups in Fig. 2. Additionally, we present the ACC of T2m for two other subsets, the first consisting of events for which two (and only two) indices among the three are strong (denoted as 2Strong+1Weak), and the second consisting of events where only one index is strong (denoted as 1Strong+2Weak). The ACC of T2m gradually increases from 0.08 to 0.24 with an increase in the number of strong coupling indices, suggesting that each metric provides at least some independent information about L-A coupling and forecast skill. It is also noteworthy that the sample sizes of the

four subsets are roughly equal. Despite the expected sample size for both the 3Strong and 3Weak groups being about 1/8 the number of the total forecasts based on the 50th percentile value, these two groups actually have larger sample sizes in the GEOS-S2S-2 dataset. This suggests that the coupling metrics within this dataset are not fully independent of each other.

We extend the analysis to week 1-4 forecasts (i.e., averaging T2m over the first four weeks of each forecast) to see if similar results apply at the monthly prediction scale. As shown on the right side of Fig. 3, the results are indeed similar – the ACC

increases consistently with the number of indices identified as strong. The difference between the 3Strong - 3Weak subsets for the week 1-4 forecasts is, however, slightly smaller than that for the 3-4 weeks forecasts (0.16 in week 3-4 forecasts versus 0.08 in week 1-4 forecasts). This reduction is likely due to the substantial influence of atmospheric initialization on skill during the first two forecast weeks.

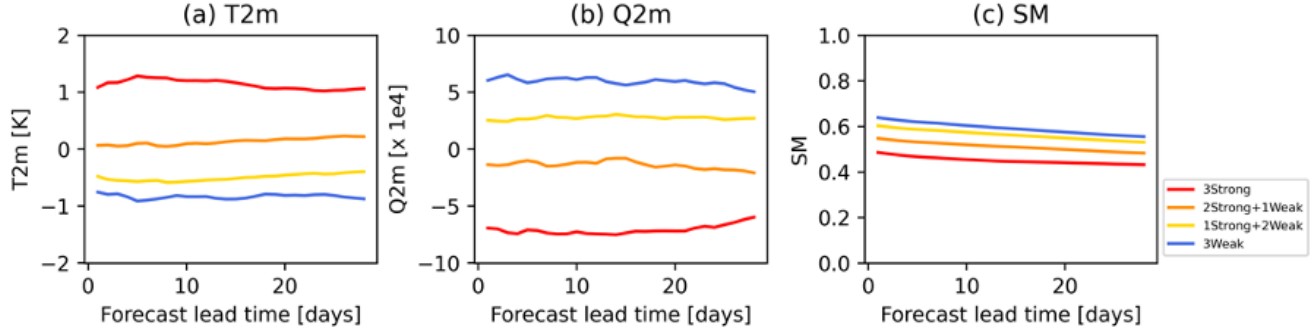

**Figure 4: Time series displaying daily-mean of (a) T2m anomalies, (b) Q2m anomalies, and (c) soil moisture averaged across the northern Great Plains (100°-85°W, 35°-45°N) for forecast lead days 1-28. The different colored lines represent the four different L-A coupling groups. Consistent with Fig. 3, red, orange, yellow, and blue lines correspond to the values in the 3Strong, 2Strong+1Weak, 1Strong+2Weak, and 3Weak groups, respectively.**

To characterize the atmospheric conditions associated with varying strengths of L-A coupling strength and prediction skill over the northern Great Plains during boreal summer, the temporal variations of areally-averaged T2m and Q2m anomalies for this region (100°-85°W, 35°-45°N) during the forecasts are depicted in Figs. 4a-b. They reveal a systematic change with an increase in the number of strong coupling indices. In cases of strong L-A coupling, warm and dry anomalies are evident in the

260 initial condition and persist for 4 weeks. In the 2Strong+1Weak group, T2m and Q2m anomalies are near zero, while the 1Strong+2Weak group shows weakly cool and humid anomalies. The coolest and most humid anomalies are seen during weak L-A coupling events.

These distinctions in T2m and Q2m anomalies can be explained, at least in part, by considerations of evaporative regimes as controlled by soil moisture. Evapotranspiration efficiency tends to increase with soil moisture in dry soil conditions (the soil
water-limited evaporative regime) and to be relatively insensitive to soil moisture changes under wet soil conditions (the energy-limited evaporative regime; e.g., Budyko, 1956, Koster et al., 2024). The rightmost panel of Fig. 4 shows that the average soil moisture gets progressively drier as the number of strong coupling indices increases. We can hypothesize that the drier soil places the evaporation into the soil water-limited evaporative regime, wherein reduced evapotranspiration induces reduced evaporative cooling. This results in warmer and drier air above the surface. Note that to the extent the evaporative
regime concept is valid, dry conditions automatically encourage high values of at least two of our coupling metrics, namely, r(SM, LH) and r(LH, TS). In some ways, it is thus not a surprise that soil water-limited conditions are associated with stronger coupling in the northern Great Plains during boreal summer in our analysis framework.

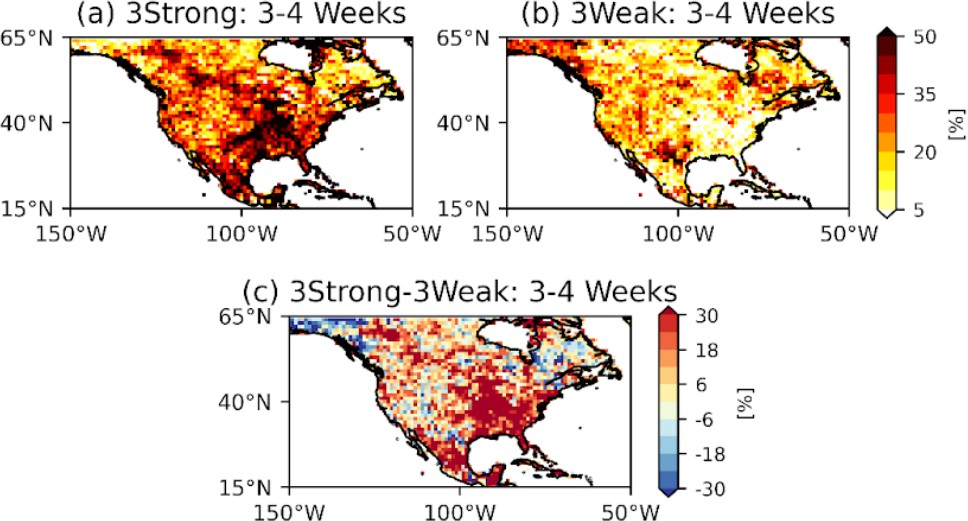

**Figure 5: Probability of predicting observed abnormally warm events during (a) 3Strong and (b) 3Weak L-A coupling events. (c) Their difference between 3Strong and 3Weak coupling events.**

Given our finding that warm and dry events are more likely during strong L-A coupling in the northern Great Plains during boreal summer, we now examine whether our coupling indices can be directly linked to a higher probability of predicting
abnormally warm events. Here we define the abnormally warm events as events for which the T2m anomalies averaged over weeks 3-4 of a forecast exceed one standard deviation of the mean of the values obtained from the 414 separate forecasts. A similar definition is applied to warm events from the ERA5 data. The probability of hitting a warm event correctly is

determined by the ratio of the number of warm events occurring in both the GEOS-S2S-2 and ERA5 data to the total number of warm events seen in the ERA5 data (Fig. 5). The 3Strong subset shows significantly higher skill than the 3Weak subset in predicting the warm events in the Midwest, northern Great Plains, and Mexico, similar to what was seen for T2m prediction (Figs. 2b-d). This suggests that our approaches and selected indices may be particularly applicable for predicting abnormally warm events during boreal summer.

## 4 Summary and Discussion

This study demonstrates that the T2m prediction skill on subseasonal timescales can be modulated by the L-A coupling strength. When strong L-A coupling (as determined by three separate coupling indices) is detected in week 3-4 forecasts, enhanced T2m prediction skill is observed across the Midwest and northern Great Plains (Fig. 2). Furthermore, the prediction skill increases with the number of indices identified as strong, suggesting that each part of the L-A coupling process from the soil to the free troposphere likely contributes to the enhancement of the prediction skill (Fig. 3). Persistent warm and dry events in the regions and season we studied here are better predicted when the L-A coupling is determined to be strong rather than weak (Figs. 4 and 5). This study suggests that the L-A coupling strength can be an indicator of stronger surface air temperature predictability in the hotspots of L-A coupling.

It is noteworthy that GEOS-S2S-2 exhibits spatially similar pattern in the L-A coupling indices with the two reanalyses (Fig. 1). However, evaluating the L-A coupling strength is challenging because the reanalysis data is influenced by the data assimilation process, which may suppress (or enhance) the L-A coupling processes modeled in the system. Long-term observational data would be needed to better evaluate the simulation and forecasting of L-A coupling.

Although Fig. 5 clearly shows that strong L-A coupling during summer in the contiguous U.S. was associated with warm anomalies, we do not directly analyze extreme events here. The L-A coupling is recognized as one of the key mechanisms driving extreme event such as heat waves and drought (e.g., Seneviratne et al. 2010). For instance, the 2012 drought in the Midwest (Roy et al. 2019) and 2022 heatwave-drought in the Great Plains (Yoon et al. 2024) were affected by the L-A coupling. Our coupling strength-based approach may help identify forecasts of opportunity for such events.

The results above are based on a single model, and L-A coupling processes are known to differ among current S2S prediction models (Abdolghafoorian and Dirmeyer, 2021). In this regard, our findings could be further evaluated by conducting a multimodel intercomparison of coupling strength impacts using the S2S prediction project data (Vitart et al., 2017) and/or SubX data (Pegion et al., 2019); such an intercomparison study would help us better understand the impact of L-A coupling over the continental U.S. in boreal summer as well as in other geographical regions and seasons.

There are various ways to construct the ensemble size for subseasonal forecasts. As the ensemble size increases, the prediction skill tends to improve (Buizza and Palmer, 1999; Vitart and Takaya, 2021). It is also valuable to examine how such adjustments to ensemble size affect the influence of the L-A coupling strength on prediction skill. We plan to investigate this further with GEOS-S2S-3, which features a larger ensemble size.

Additionally, seasonal prediction models have exhibited systematically warm and dry biases over the central U.S. (Klein et al., 2006; Ardilouze et al., 2019). The impacts of such biases on forecast skill are complex; Koster et al. (2021), for example, found that a precipitation bias has a distinctly different impact if the soil starts out anomalously wet rather than anomalously dry. The impacts of such biases on L-A coupling will need to be further investigated in future studies. Through multimodal intercomparison, we can better understand the possible impacts of these biases on L-A coupling and predictability.

## Acknowledgements

This work was funded by the NASA S2SHYD (NNH21ZDA001N-S2SHYD) program. The GEOS-S2S-2 forecasts and the MERRA-2 reanalysis were generated by NASA's Global Modeling and Assimilation Office.

## Competing interests

The contact author has declared that none of the authors has any competing interests.

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
