# Peer review of "The role of land-atmosphere coupling in subseasonal surface air temperature prediction across the contiguous United States"

_EGUsphere, 2024_

## Referee Comment (RC2)

**General comments**

The authors present a thorough evaluation of land-atmosphere (L-A) coupling and its influence on subseasonal-to-seasonal (S2S) predictability, specifically focusing on surface air temperature (T2m). They assess the coupling processes by calculating correlations that capture the interactions between land conditions and surface fluxes, as well as between surface fluxes and the overlying atmosphere or boundary layer. The study analyzes 414 forecast dates using Version 2 of NASA's advanced GEOS S2S analysis and forecast system, with comparisons to ERA5 and MERRA2 reanalyses. The authors effectively emphasize the importance of understanding L-A coupling strength for enhancing forecast skill, especially for "forecasts of opportunity" across the continetal U.S.

Your study provides valuable insights into the role of strong L-A coupling for enhancing surface air temperature prediction on subseasonal-to-seasonal timescales. One area that could enrich this research further would be an exploration of how this coupling influences the predictability of specific extreme events, such as heatwaves and soil moisture droughts, or their compound occurrences. Understanding these interactions could provide additional context for the societal and ecological impacts of such events and improve risk management strategies. Do you see potential in integrating analyses of compound heatwave-drought events with your L-A coupling framework to advance this aspect?

Below, I offer several suggestions to strengthen the paper.

**Specefic Comments**

**Title**
The current title, "The role of land-atmosphere coupling in subseasonal surface air temperature prediction," could be refined to better reflect the study's specific focus. I suggest including the geographic focus and indicating the use of the GEOS-S2S-2 model and reanalysis data for increased specificity.

**Introduction**
1.  The introduction could benefit from integrating recent studies that highlight how the development of compound drought-heatwave events is influenced by distinct land-atmosphere (L-A) coupling behaviors associated with water and energy limitation regimes in various U.S. regions. For instance, the work by Yoon et al. (2024) emphasizes how L-A coupling significantly shaped the 2022 compound drought-heatwave events in the contiguous United States (CONUS). Incorporating these findings could provide context on how specific L-A coupling dynamics contribute to such extreme events and enhance the relevance of your study's focus on prediction skill. This perspective would enrich the discussion by linking L-A coupling to practical forecasting challenges and implications.

**Data and Methods**

1. **Geographical Focus**: While the study seems to focus on the continental U.S., explicitly stating this in the Data and Methods section would help clarify the study's regional scope for readers.

2. **Model and Data Source Clarity**: The authors use the NASA GEOS-S2S model from 1999 to 2022 and compare its outputs to ERA5 and MERRA2 reanalyses. It would be beneficial if the authors could specify the strengths and limitations of each reanalysis data source in the context of L-A coupling.

3. **Forecast Initialization Skill (Lines 52-54)**: The paper emphasizes forecasts of coupling components rather than initialization. Including a brief analysis of GEOS-S2S forecast skill compared to reanalysis data specifically for weeks 3-4 could be valuable for readers assessing the model's performance.

4. **Order Consistency**: In the Methods section, the authors first introduce the Anomaly Correlation Coefficient (ACC) before discussing L-A coupling metrics. However, this order is not followed in the Results section. For consistency and readability, aligning the order in both sections would be beneficial.

**Results**

1. **Figure 1 (a-c)**: The spatial similarity between GEOS-S2S and the two reanalyses is noteworthy. Expanding on these similarities in the discussion would strengthen the analysis and provide further context for the comparison.

2. **Figure 2 (d-e)**: The correlation between latent heat flux (LH) and surface skin temperature (TS) appears stronger in ERA5 than in GEOS-S2S in the western U.S. Providing an explanation for this difference could offer insights into the model's performance in capturing regional variations.

3. **Figure 2 (g-i)**: The spatial pattern for the correlation between LH and LCLd in GEOS-S2S appears more similar to that of MERRA2. The authors could address the inconsistencies observed when comparing GEOS-S2S with the two reanalyses in Figures 2(d-e) and (g-i) to enhance the interpretation of these results.

**Summary and Discussion**

1. **Multimodel Comparison**: While the study provides useful insights based on the GEOS-S2S model, the results could be more generalizable if tested across multiple models. I agree with the authors' suggestion for a multimodel intercomparison using S2S project data or SubX data, which could broaden the understanding of L-A coupling impacts across diverse geographical and seasonal contexts.

2. **Biases in Seasonal Models**: The authors mention the presence of systematic warm and dry biases in certain seasonal prediction models over the central U.S. (e.g., Klein et al., 2006; Ardilouze et al., 2019). Expanding the discussion on how these biases may impact L-A coupling strength and forecast skill could provide additional context and depth.

3. **Ensemble Size and Forecast Frequency**: Previous studies indicate that increasing forecast frequency can enhance S2S forecast accuracy. Similarly, expanding ensemble size may better capture subtle shifts in forecast probabilities (e.g., see ECMWF newsletter 173). Including a discussion on the potential impacts of ensemble size and forecast frequency would add depth and highlight possible avenues for future studies.

4. **Incorporating Examples of Extreme Events:** Integrating specific examples, such as the 2011 Southern Plains Drought and Heatwave, the 2012 Central U.S. Drought and Heatwave, and the 2020–2022 Western U.S. Megadrought and associated heatwaves, would contextualize the role of L-A coupling dynamics in extreme events. These cases illustrate the compound impacts of heat and drought and how L-A coupling can amplify such conditions. Addressing these examples could enhance the relevance of the study by connecting findings to real-world events. Additionally, discussing these interactions in the 'Discussion and Conclusions' section would provide insights into how future research could explore these dynamics to improve forecasting for compound events.

**References**

Vitart, F., Balmaseda, M. A., Ferranti, L., & Fuentes, M. (2022). The Next Extended-Range Configuration for IFS Cycle 48r1. *ECMWF Newsletter*, no. 173 (Autumn 2022). Retrieved from https://www.ecmwf.int/en/newsletter/173/news/next-extended-range-configuration-ifs-cycle-48r1.

Yoon, D., Chen, J. H., & Seo, E. (2024). Contribution of land-atmosphere coupling in 2022 CONUS compound drought-heatwave events and implications for forecasting. *Weather and Climate Extremes*, *46*, 100722.https://doi.org/10.1016/j.wace.2024.100722.

---

## Author Comment (AC1)

The study investigates predictions of surface air temperature under strong and weak L-A coupling conditions using NASA's GEOS S2S forecast system. The authors assess the predictive skill of temperature forecasts during weeks 3 and 4 of the boreal summer, particularly in the Midwest and northern Great Plains, by applying L-A coupling metrics. These metrics capture the relationships between soil moisture, latent heat flux, and some other atmospheric variables, such as surface skin temperature and planetary boundary layer height. The study hypothesizes that surface temperature predictions improve when strong coupling is present. The results show that strong L-A coupling increases the ability to predict extreme warm events, especially for the northern Great Plains, where strong coupling increases the likelihood of correctly predicting abnormally warm temperatures during weeks 3-4.

While the paper is well-written and concise, the discussions are limited and could be expanded to address several key issues. For example, it should be addressed how some preprocessing explained in the method section, such as upscaling or spatiotemporal aggregation of the different datasets to match each other, plays a role in the results. Moreover, the strength of land-atmosphere coupling is mostly mediated by soil moisture, especially in water-limited regions. Despite its importance, there are no discussions about the accuracy of the soil moisture used in this study. Even a small bias in soil moisture values used in this study, especially in heavily irrigated regions during the growing season, may have a significant impact on the subseasonal air temperature predictions. Providing a comparison with observational-based soil moisture observations such as SMAP would more clearly identify the regions where the strength of land-atmosphere coupling is more reliable in contributing to air temperature prediction skills.

Thank you for your comments. Below we address your two key points: 1) the impact of the spatial interpolation on our results and 2) the accuracy of the soil moisture.

1) The impact of spatial interpolation

Motivated by your comment, we conducted the same analysis using GEOS-S2S-2 data and ERA5 on a 0.5°x0.5° grid (without spatial interpolation for GEOS-S2S-2). Result, presented in Fig. R1, are similar to what we found in the original analysis. Accordingly, we have added the following sentence at line 218:

"To test whether our spatial interpolation of the GEOS-S2S-2 data had some impact on our findings, we conducted the same analysis (not shown) using a 0.5°x0.5° grid without the spatial interpolation. The results were essentially the same."

[Figure]

**Figure R1**. The ACC of T2m anomalies (forecast values versus observations) at a forecast lead time of week 3-4 over North America during (a) all, (b) strong, and (c) weak L-A coupling events. Strong (weak) events are defined when all three indices are above (below) the 50 percentiles. (d) The difference in T2m ACC between strong and weak events. Statistically significant differences at a 90% confidence level are shaded.

2) The accuracy of the soil moisture

We agree with the reviewer; even a small bias in soil moisture values used in this study, especially in heavily irrigated regions during the growing season, may have a significant impact on the subseasonal air temperature predictions. As background for the manuscript's new text about this, we should point out that there is no fully satisfactory way of evaluating the soil moisture accuracy. SMAP data exist from only from 2015 onward, allowing for only a small overlap period; furthermore, like all satellite-based products, the SMAP data have their own biases. Assimilation-based soil moisture datasets (GLDAS, NLDAS, etc.) are model-dependent and thus subject to model assumptions, and the in-situ measurement sites that provide soil moisture are point measurements that, due to the spatial representativeness problem, may not accurately describe the grid-scale soil moistures that our study relies on.

All this being said, we have added the following text to the paper at line 76 in response to the reviewer's comment:

"The weakly coupled data assimilation system used for the forecast system's initialization makes use of observed (rain gauge-based) precipitation measurements to drive the land surface fields; as a result, the soil moisture initialization for the forecasts appropriately reflects the character of the antecedent observed precipitation. Quantitative estimates of the accuracy of such precipitation-driven, model-based estimates are rare but do exist. Using a slightly modified version of the land model used here in GEOS (Catchment model), Reichle et al. (2017) found strong temporal correlations between the surface and root zone soil moisture so generated and corresponding in-situ measurements across the contiguous US (CONUS). The soil moistures produced by a slightly updated version of the land model were recently compared to Soil Moisture Active Passive (SMAP) satellite-based soil moisture retrievals (Entekhabi et al., 2009); the resulting anomaly correlation coefficients over CONUS were found to generally exceed 0.6 (Qing Liu, personal comm., 2024).

Koster et al. (2020, their Figure 9) effectively show that significant subseasonal air temperature forecast skill in the GEOS-S2S-2 system, particularly in the eastern half of CONUS, is strongly tied to the initialized profile soil moisture, indicating useful accuracy therein."

---

## Author Comment (AC2)

**General comments**

The authors present a thorough evaluation of land-atmosphere (L-A) coupling and its influence on subseasonal-to-seasonal (S2S) predictability, specifically focusing on surface air temperature (T2m). They assess the coupling processes by calculating correlations that capture the interactions between land conditions and surface fluxes, as well as between surface fluxes and the overlying atmosphere or boundary layer. The study analyzes 414 forecast dates using Version 2 of NASA's advanced GEOS S2S analysis and forecast system, with comparisons to ERA5 and MERRA2 reanalyses. The authors effectively emphasize the importance of understanding L-A coupling strength for enhancing forecast skill, especially for "forecasts of opportunity" across the continental U.S.

Your study provides valuable insights into the role of strong L-A coupling for enhancing surface air temperature prediction on subseasonal-to-seasonal timescales. One area that could enrich this research further would be an exploration of how this coupling influences the predictability of specific extreme events, such as heatwaves and soil moisture droughts, or their compound occurrences. Understanding these interactions could provide additional context for the societal and ecological impacts of such events and improve risk management strategies. Do you see potential in integrating analyses of compound heatwave-drought events with your L-A coupling framework to advance this aspect?

⇒ We agree with the reviewer's comment in terms of the broader potential applicability and impact of the study and approach. The focus of the current study is on the impact of L-A coupling on prediction skill of T2m as a whole, rather than on the prediction skill of individual extreme events that comprise a subset of this bulk analysis. In response to the reviewer's suggestion, we have added a discussion on the potential utility of this approach for improving forecasts of opportunity for extreme events. However, an actual examination of extreme events is beyond the scope of this study.

**Specific Comments**

**Title**

The current title, "The role of land-atmosphere coupling in subseasonal surface air temperature prediction," could be refined to better reflect the study's specific focus. I suggest including the geographic focus and indicating the use of the GEOS-S2S-2 model and reanalysis data for increased specificity.

⇒ Following the reviewer's comment, we have included the geographic focus in the title to enhance clarity. The revised title is now: "The role of land-atmosphere coupling in subseasonal surface air temperature prediction across the contiguous United States."

**Introduction**

1.  The introduction could benefit from integrating recent studies that highlight how the development of compound drought-heatwave events is influenced by distinct land-atmosphere (L-A) coupling behaviors associated with water and energy limitation regimes in various U.S. regions. For instance, the work by Yoon et al. (2024) emphasizes how L-A coupling significantly shaped the 2022 compound drought-heatwave events in the contiguous United States (CONUS). Incorporating these findings could provide context on how specific L-A coupling dynamics contribute to such extreme events and enhance the relevance of your study's focus on prediction skill. This perspective would enrich the discussion by linking L-A coupling to practical forecasting challenges and implications.

    ⇒ Thank you for your suggestion. Our focus is on the impact of L-A coupling on prediction skill of T2m rather than on the prediction skill of specific extreme events. Therefore, instead of adding such

discussion to the introduction, we have included at lines 301-305 a discussion about the potential relevance of our approach to predicting extreme events. This also addresses the reviewer's comment 4 below in the Summary and Discussion section. The new text is as follows:

"Although Fig. 5 clearly shows that strong L-A coupling during summer in the contiguous U.S. was associated with warm anomalies, we do not directly analyze extreme events here. The L-A coupling is recognized as one of the key mechanisms driving extreme event such as heat waves and drought (e.g., Seneviratne et al. 2010). For instance, the 2012 drought in the Midwest (Roy et al. 2019) and 2022 heatwave-drought in the Great Plains (Yoon et al. 2024) were affected by the L-A coupling. Our coupling strength-based approach may help identify forecasts of opportunity for such events."

**Data and Methods**

1. **Geographical Focus**: While the study seems to focus on the continental U.S., explicitly stating this in the Data and Methods section would help clarify the study's regional scope for readers.

    $\Rightarrow$ We have added a sentence to clarify the geographic focus in introduction at line 52 as shown below:

    "The analysis focuses on the contiguous United States, which contains a well-known hotspot of L-A coupling (e.g., Koster et al. 2006)."

2. **Model and Data Source Clarity**: The authors use the NASA GEOS-S2S model from 1999 to 2022 and compare its outputs to ERA5 and MERRA2 reanalyses. It would be beneficial if the authors could specify the strengths and limitations of each reanalysis data source in the context of L-A coupling.

    $\Rightarrow$ The advantage of using MERRA-2 as reference data is its ability to account for the data assimilation effect, as noted in lines 181-185. However, MERRA-2 incorporates no surface temperature information over land during the data assimilation process. We mentioned this in the manuscript, but we revised the manuscript on lines 173-179 to clarify this point, as shown below.

    "The spatial distribution of GEOS-S2S-2 is understandably more similar to that of MERRA-2, as both are based on the same modeling system (Molod et al. 2020). The asymmetry between the western and eastern U.S. is somewhat more pronounced in ERA5 compared to GEOS-S2S-2 and MERRA-2. As mentioned earlier, the data assimilation process used in the production of ERA5 incorporated substantial amounts of T2m information from ground stations. Furthermore, ERA5 is based on higher spatial resolution: ERA5 data are provided at approximately 25 km horizontal resolution, whereas MERRA-2 provides at 50 km. This difference may contribute to variations in simulating boundary layer conditions and land-atmosphere coupling. Nevertheless, using MERRA-2 is still valuable for examining the sensitivity of the results to the choice of reanalysis data."

3. **Forecast Initialization Skill (Lines 52-54)**: The paper emphasizes forecasts of coupling components rather than initialization. Including a brief analysis of GEOS-S2S forecast skill compared to reanalysis data specifically for weeks 3-4 could be valuable for readers assessing the model's performance.

    $\Rightarrow$ We do not fully understand the reviewer's suggestion. If the reviewer is asking us to provide the prediction skill of T2m, we have already presented it in Fig. 2a. However, if the reviewer is requesting the prediction skill of coupling indices, this is challenging to address due to the influence of data assimilation. As mentioned in lines 181-183, the data assimilation process disrupts L-A coupling, suppressing or enhancing the physical correlations in the reanalysis data. In response to reviewer, we have expanded this point at lines 298-300 in the discussion section as shown below:

"However, evaluating the L-A coupling strength is challenging because the reanalysis data is influenced by the data assimilation process, which may suppress (or enhance) the L-A coupling processes modeled in the system. Long-term observational data would be needed to better evaluate the simulation and forecasting of L-A coupling."

4. **Order Consistency**: In the Methods section, the authors first introduce the Anomaly Correlation Coefficient (ACC) before discussing L-A coupling metrics. However, this order is not followed in the Results section. For consistency and readability, aligning the order in both sections would be beneficial.

⇒ We have switched the order of Sections 2.3 and 2.4 as the reviewer suggested.

**Results**

1. **Figure 1 (a-c)**: The spatial similarity between GEOS-S2S and the two reanalyses is noteworthy. Expanding on these similarities in the discussion would strengthen the analysis and provide further context for the comparison.

⇒ We believe that the spatial similarity has already been described in detail in lines 151-166. While we are unsure if we have fully understood the reviewer's suggestion, we now revisit this point in the discussion section at line 297:

"It is noteworthy that GEOS-S2S-2 exhibits spatially similar pattern in the L-A coupling indices with the two reanalyses (Fig. 1)."

1. **Figure 1 (d-e)**: The correlation between latent heat flux (LH) and surface skin temperature (TS) appears stronger in ERA5 than in GEOS-S2S in the western U.S. Providing an explanation for this difference could offer insights into the model's performance in capturing regional variations.

⇒ Thank you for the suggestion. The additional text on lines 173-179 to address this reviewer's comment 2 in Data and Methods section provides a partial answer to this question. As to the explanation, while we cannot be certain, we speculate that it may be due to data assimilation or differences in the land model. Specifically, ERA5 assimilates 2m T and q, and uses this information to nudge soil moisture in order to influence the surface fluxes and energy balance to better match the 2m observations. This creates a tight link and constraint on the surface based on the 2m observations that the other models and reanalyses do not enforce.

2. **Figure 1 (g-i)**: The spatial pattern for the correlation between LH and LCLd in GEOS-S2S appears more similar to that of MERRA2. The authors could address the inconsistencies observed when comparing GEOS-S2S with the two reanalyses in Figures 1(d-e) and (g-i) to enhance the interpretation of these results.

⇒ Yes. The spatial pattern in GEOS-S2S-2 is more similar to that of MERRA-2 because they utilize a fundamentally similar model and physical parameterizations (Molod et al. 2020). This is pointed out in the new text at lines 173 forward, as discussed in the responses to earlier comments. We also mention in that new text that the inconsistences seen when comparing GEOS-S2S-2 with the ERA5 may stem from the assimilation in the latter of T2m data.

**Summary and Discussion**

1. **Multimodel Comparison**: While the study provides useful insights based on the GEOS-S2S model, the results could be more generalizable if tested across multiple models. I agree with the authors' suggestion for a multimodel intercomparison using S2S project data or SubX data, which could broaden the understanding of L-A coupling impacts across diverse geographical and seasonal contexts.

   ⇒ Thank you for agreeing to it. We will examine it in the next study.

2. **Biases in Seasonal Models**: The authors mention the presence of systematic warm and dry biases in certain seasonal prediction models over the central U.S. (e.g., Klein et al., 2006; Ardilouze et al., 2019). Expanding the discussion on how these biases may impact L-A coupling strength and forecast skill could provide additional context and depth.

   ⇒ The study of Koster et al. (2021), found, for example, that the bias has a distinctively different impact in wet and dry regimes, highlighting the complexities of the bias and so its potential impact on skill. The impacts on our results will need to be investigated further in future studies. We have added this discussion point at line 315:

   "Additionally, seasonal prediction models have exhibited systematically warm and dry biases over the central U.S. (Klein et al., 2006; Ardilouze et al., 2019). The impacts of such biases on forecast skill are complex; Koster et al. (2021), for example, found that a precipitation bias has a distinctly different impact if the soil starts out anomalously wet rather than anomalously dry. The impacts of such biases on L-A coupling will need to be further investigated in future studies."

3. **Ensemble Size and Forecast Frequency**: Previous studies indicate that increasing forecast frequency can enhance S2S forecast accuracy. Similarly, expanding ensemble size may better capture subtle shifts in forecast probabilities (e.g., see ECMWF newsletter 173). Including a discussion on the potential impacts of ensemble size and forecast frequency would add depth and highlight possible avenues for future studies.

   ⇒ Thank you for suggesting it. It is helpful in improving our discussion. We acknowledge the importance of a larger number of ensemble size, which under certain conditions contributes to improved forecast skill. It is worth examining this in the future, and we have added the following discussion at lines 311-314:

   "There are various ways to construct the ensemble size for subseasonal forecasts. As the ensemble size increases, the prediction skill tends to improve (Buizza and Palmer, 1999; Vitart and Takaya, 2021). It is also valuable to examine how such adjustments to ensemble size affect the influence of the L-A coupling strength on prediction skill. We plan to investigate this further with GEOS-S2S-3, which features a larger ensemble size."

4. **Incorporating Examples of Extreme Events:** Integrating specific examples, such as the 2011 Southern Plains Drought and Heatwave, the 2012 Central U.S. Drought and Heatwave, and the 2020–2022 Western U.S. Megadrought and associated heatwaves, would contextualize the role of L-A coupling dynamics in extreme events. These cases illustrate the compound impacts of heat and drought and how L-A coupling can amplify such conditions. **Addressing these examples could enhance the relevance of the study by connecting findings to real-world events.** Additionally, **discussing these interactions** in the 'Discussion and Conclusions' section **would provide insights into how future research could explore these dynamics to improve forecasting for compound events.**

$\Rightarrow$ Thank you for the suggestion. We have addressed this in the revised manuscript (lines 301-305) as outlined in our response to the previous comment related to the Introduction.

---

## Author Response (AR2)

**Response to Reviewer 2**

**Comments:** The authors have made major revisions to the manuscript, addressing most of the reviewers' comments. These improvements include a more thorough discussion of the study's limitations and the restructuring of sections as requested.

One aspect that remains unaddressed is the specificity of the title. In the initial submission, the study region was not included, but the authors appropriately added this in the revision. However, it was also recommended that the title explicitly reflect the model used for greater clarity. This recommendation was not addressed in their response. Since the study is based on a single model (GEOS-S2S) and its comparison with two reanalysis datasets, the phrase "The Role of" in the title may be too strong, as it suggests a more generalizable conclusion about land-atmosphere coupling in subseasonal prediction across the contiguous United States. Given that the study does not employ a multi-model assessment (which authors have already referred to in lines 305-39), a more precise wording would better reflect the study's scope and avoid potential overgeneralization. I still suggest refining the title, as this would enhance precision and better align it with the study's scope.

Overall, the authors have improved the clarity and depth of the manuscript. However, refining the title would further enhance its precision and ensure it accurately reflects the study's focus.

**Response:** Thank you for your comments. We explored options to expand the title to clarify that this study is based on a single modeling system. However, every attempt resulted in a title that felt overly long and cumbersome. Given that the abstract clearly states the use of a single modeling system, we believe the current title remains the most effective. Therefore, if the editor agrees, we would like to retain the title as it is.